# Effect of Supergravity Field on the Microstructure and Mechanical Properties of Highly Conductive Cu Alloys

**DOI:** 10.3390/ma16062485

**Published:** 2023-03-21

**Authors:** Lu Wang, Xi Lan, Zhe Wang, Zhancheng Guo

**Affiliations:** State Key Laboratory of Advanced Metallurgy, University of Science and Technology Beijing, Beijing 100083, China

**Keywords:** supergravity field (SGF), grain refining mechanism, tensile properties, Cu-Sn alloy

## Abstract

In consideration of the characteristics of supergravity to strengthen solidification structures, the effect of the supergravity field (SGF) on the grain refinement and mechanical properties of Cu-0.5Sn alloys was investigated in this paper. Firstly, it was experimentally verified that the addition of Sn could effectively refine the grain. Subsequently, the variations in grain size, tensile strength, and plasticity of the Cu-0.5Sn alloy were compared in normal and SGF conditions. The results revealed that the tensile strength and plasticity of the alloy increased with the increase in gravity coefficient. The ultimate tensile strength of the Cu-0.5Sn alloy in a normal gravity field was 145.2 MPa, while it was 160.2, 165.3, 167.9, and 182.0 MPa in an SGF with G = 100, 300, 500, and 1000, respectively, and there was almost no effect on conductivity. Finally, it was clarified that the mechanism of grain refinement by SGF was that the intense convection caused the fracture of the dendrites to become new nucleating particles. The increased viscosity under SGF hindered the diffusion of atoms in the melt and slowed down the movement of atoms toward the nucleus, leading to a decrease in grain size.

## 1. Introduction

Copper (Cu) alloys with outstanding electrical conductivity, high strength, and enough ductility are of considerable interest in the electronics, transportation, and aerospace sectors, including electrical contact materials and high-speed rail contact wires [1,2,3,4]. However, it is difficult to improve both electrical conductivity and strength at the same time due to the contradictory properties of Cu conductors [5,6]. The trade-off between strength and conductivity has persisted for a long time in the study of high-strength and high-conductivity (HSHC) Cu materials, as well as in the entire field of conductive material research [7]. HSHC Cu alloys can be strengthened using a variety of processes, the most common of which are solid solution strengthening [8,9], fine grain strengthening [10,11], precipitation strengthening [12,13], and deformation strengthening [11,14,15]. However, these strengthening methods are susceptible to causing a large number of minute crystal defects, which result in the lower conductivity of alloys [1,3,4,16]. How to attain the HSHC of Cu, i.e., to compromise the conductivity of Cu as little as possible, while substantially increasing its strength, is an essential subject. It is well known that control of the solidification structure and defects in the initial cast ingots of the alloy is crucial for the mechanical properties and has a significant impact on the subsequent heat treatment, rolling processes, and the final group morphology. Therefore, the as-cast solidification structure and defects of Cu alloys are investigated first in this paper.

The presence of convection in molten metals has a significant impact on the formation of structure and defects [17]. For instance, the unreasonable control of heat transfer conditions, as well as the metal’s solidification shrinkage characteristics, often result in the formation of defects such as pores in the sample. Physical field treatment technologies can greatly benefit the solidification process of metals by refining the grains, improving the solidification structure, reducing the segregation defects, etc., ultimately enhancing the comprehensive performance of castings. Currently, several intensifying techniques such as magnetic [18,19], electric current [20], and ultrasonic [21] fields have been proposed for application in grain refinement, but they are generally studied in the field of solidification. The supergravity technique is a more successful intensification technology than the preceding approaches are, and the current research on supergravity solidification mainly focuses on melt purification, microstructure segregation, and grain refinement [22,23,24,25,26]. Although the solidification behavior in the SGF and the available research results have attracted attention, the grain refining of castings in the SGF has been the subject of only a few comprehensive investigations [27]. Research on the supergravity solidification mechanism remains controversial.

The mechanism of supergravity solidification has been explained differently in various reports. For example, Chen et al. [28] concluded that the structure of the as-cast alloys could be refined under SGF (centrifugal force) because the addition of supergravity increased the driving force of Gibbs free energy for nucleation, thereby reducing the critical nucleation radius and increasing the nucleation rate. Zhao et al. [26] reported that the main mechanism of grain refinement by supergravity was to break dendrites and cause heterogeneous nucleation, with the change in nucleation energy caused by supergravity having a minimal effect on grain refinement. Yang et al. [25] analyzed the microstructures of supergravity solidification and found that supergravity can promote nucleation and refine grains only when it is applied at the initial solidification stage, thus proposing the “crystal rain” mechanism. Li et al. [29] investigated the grain refinement mechanism of Al-Zn-Li-Mg-Cu entropic alloys in the SGF. The results indicated that the refiners refined grains due to the variation in pressure and viscosity under the SGF. Although it has been more and more experimentally demonstrated that supergravity can refine the solidified structure, the experimental conditions in different studies have varied greatly. Furthermore, there have been no reports on supergravity solidification in high-conductivity Cu alloys containing trace alloying elements (within the solid solubility).

In light of this, this study aims to investigate the effect of supergravity on the grain size and mechanical properties of Cu-Sn alloy castings using the solidification with centrifugation method. Moreover, the mechanism of the supergravity effect on grain refinement was also analyzed in detail.

## 2. Experimental

### 2.1. Experimental Equipment

The supergravity apparatus consists of two furnaces (the heating furnace and the counterweight furnace) revolving around a rotation axis, as depicted in the schematic in Figure 1. A programable controller and B-type thermocouple are used to control the heating furnace temperature (within ±3 °C). In addition, the slip ring attached to the centrifugal axis enables power transfer and electrical signaling during rotation. Before each experiment, the center of gravity of the apparatus first needs to be carefully adjusted to maintain balance.

The gravity coefficient (G) is determined as the ratio of centrifugal acceleration to normal gravity acceleration to evaluate the SGF (as shown in Equation (1)).
(1)$$G=\frac{\sqrt{g^{2}+{\left(\omega^{2}r\right)}^{2}}}{g}=\frac{\sqrt{g^{2}+{\left(\frac{N^{2}\pi^{2}r}{900}\right)}^{2}}}{g}$$where *ω* is the angular velocity (in rad·s^−1^), *N* is the rotating speed of the centrifuge (in r·min^−1^), and *r* is the distance from the axis to the sample center (equal to 0.25 m in this work).

### 2.2. Materials Preparation

The raw materials of the Cu-(0, 0.3, and 0.5) Sn alloys (mass fraction, %) were melted in a vacuum induction melting furnace under an argon atmosphere using a mixture of highly pure Cu and Sn (99.99wt%). The pure Cu and Sn were placed in a graphite crucible and heated to 1200 °C under argon gas (99.99%) protection and held at this temperature for 1 h. The ingots were remelted several times to ensure homogenization, and then about 20 g of cylinders (20 mm in diameter) were cut using a wire electrode for further experiments. A mixture of sodium chloride (45wt%) and potassium chloride (55wt%) was used as a covering slag to protect from the oxidation of samples during the supergravity experiments.

### 2.3. Experimental Procedure

In total, 110 g of Cu-0.5Sn alloy metal and 10 g of covering slag were placed in a quartz crucible (20 mm inner diameter), which was then put into a graphite crucible with a lid (27 mm inner diameter), followed by heating at 1200 °C in the supergravity heating furnace for 10 min. After heat preservation, the supergravity apparatus was initiated and adjusted to target the rotating speed. Under the SGF, the temperature of the melt was set to drop at a rate of 10 °C·min^−1^. To ensure full solidification of the sample, the supergravity apparatus was operated until the temperature decreased to 900 °C, at which point, it was turned off, and the sample was promptly quenched in water to preserve its current structure. The following values are the gravity coefficients of studies 1, 100, 300, 500, and 1000, with the corresponding rotating speeds of 0, 598, 1036, 1338, and 1892 r·min^−1^, respectively.

The obtained samples were cut in half along the direction of the SGF, and the cutting face was burnished, polished, and then etched with a solution of 5 g FeCl_3_ + 50 mL HCl + 100 mL H_2_O to reveal the structure of the investigated alloys. The grain size of the alloy was obtained by analyzing the images using Image-Pro Plus software. The microstructures of the alloy were analyzed through electron backscatter diffraction (EBSD) using the Symmetry S2 equipment. The electrical conductivity was measured at room temperature using an electrical conductivity tester (AT 510). Using an electronic universal testing machine (WDW-10e), tensile tests were conducted to determine the influence of SGF on the tensile characteristics of the Cu-0.5Sn alloy. The tensile tests were conducted at room temperature at an initial displacement rate of 0.5 mm·min^−1^, and the specimen is shown in Figure 2.

## 3. Results and Discussion

### 3.1. Effect of Sn Content on the Microstructure and Properties of Cu-Sn Alloys

To investigate the effects of Sn content on the solidification structures of refining Cu alloys, experiments were performed using three distinct alloys with Sn weight percentages of 0, 0.3, 0.5, and 0.7, respectively. The Cu alloys were obtained in a normal gravity field at a 10 °C·min^−1^ cooling rate; as shown in Figure 3, the first line (G = 1) corresponds to Sn contents of 0wt%, 0.3wt%, 0.5wt%, and 0.7wt%, respectively. It can be clearly seen that there was no obvious grain morphology in pure Cu due to the large grain size (Figure 3). With the increase in Sn content to 0.3wt%, the structure exhibited coarse and irregularly shaped grains. Upon further increasing the Sn content to 0.7wt%, the grain size of the samples became more refined. That is to say, the Sn addition played a role in promoting the refinement of grains, which was caused by the fact that as the Sn content increased, both the number of nuclei per unit volume and the nucleation rate increased, and the room available for the growth of each grain became smaller. In a longitudinal comparison, the grain size obviously decreased with the increase in the gravity coefficient. The grains gradually changed from coarse grains to equiaxed grains. Furthermore, it can be seen that the grain refinement effect of SGF was obviously enhanced by the increasing solute Sn concentration. This was due to the fact that the rate of heterogeneous nucleation was increased during the solidification process with the increasing solute Sn content, and dendrites grew much faster. Meanwhile, the convection in the alloy was enhanced by the increased supergravity coefficient. During the process of grain growth, it was easy to be broken to produce new fragments to become a new nucleation. Hence, the enhancement of grain refinement was caused by the increase in solute content in the SGF.

Figure 4 shows the stress–strain curves of Cu alloys with distinct Sn contents, demonstrating that the introduction of Sn increased the strength of pure Cu while reducing its plasticity. This was because the addition of 0.3wt%, 0.5wt%, and 0.7wt% Sn to the Cu matrix metal led to the formation of a solid solution, thereby causing lattice distortion. The interaction between lattice distortion and the elastic field of dislocations restricted the movement of dislocations on the slip plane, resulting in a sharp increase in the strength of Cu-Sn alloys. However, its plasticity was lower than the average value of pure metal. Additionally, the incorporation of Sn atoms led to a significant decrease in alloy conductivity. Because Sn atoms were dissolved in the Cu matrix lattice, the lattice distortion of Cu destroyed the periodicity of the lattice potential field, which increased both the probability of electron scattering and the resistance. Table 1 summarizes the specific experimental outcomes.

### 3.2. Effect of Supergravity on the Microstructure and Properties of Cu-Sn Alloys

The above results revealed that the incorporation of Sn elements caused the solid solution strengthening to increase the alloy strength, but plasticity and electrical conductivity decreased significantly. Grain refining is known to increase both the strength and plasticity of alloys. Previous studies indicated that supergravity plays a role in grain refinement. Therefore, the influence of different gravity coefficients on the mechanical properties and electrical conductivity of Cu-0.5Sn was investigated in this paper.

Figure 5 presents the vertical sections of the samples obtained in the gravity coefficient range of 1–1000 and with the temperature decreased from 1200 °C to 900 °C. The solidification structure was mainly composed of coarse crystals in the condition of G = 1, as exhibited in Figure 5a. The solidification structure of Cu-0.5Sn alloy resembled that of a normal gravity field at G = 100, as shown in Figure 5b, albeit with a more uniform crystal size. This was because the convection of the melt increased under supergravity and the melt dissipated heat more uniformly during solidification. However, the convection of the alloy melt caused by the gravitational field under the low gravity coefficient could not overcome the intermolecular forces to produce dendrite fracture. Therefore, the grain size of the alloy was relatively uniform, but it still appeared to be coarse-grained. When the gravity coefficient increased to 300 and 500, the fine grains occupied most of the sample, as shown in Figure 5c,d. Upon further increasing the gravity coefficient to 1000, the grain size of the Cu-0.5Sn alloy was reduced from 4.3 mm to 0.8 mm, and the structure of the alloy was denser and transformed into more uniform equiaxed crystals from coarse crystals. This was due to enhanced convection and heat exchange in the alloy as the gravity coefficient increased. During the solidification process, the breaking of the dendrites created new nucleating particles, so that the grains were refined. Simultaneously, the melt heat dissipation lost its directionality, allowing the nucleus of the crystal to grow freely in the liquid at a rate that was almost the same in all directions. As a result, the shape of the crystals gradually transformed into equiaxed crystals.

To more intuitively characterize the effect of different gravity coefficients on the grain size of the alloy, grain size statistical analysis of the samples was carried out using Image-Pro Plus software, as summarized in Figure 6. The average grain size was 4.3 mm in the normal gravity field. With the increased gravity coefficient of G = 100, the average grain size was reduced to 3.2 mm. The grain size further decreased to just 0.8 mm, and the gravity coefficient was up to 1000. Thus, the average grain size decreased as the gravity coefficient increased, which is consistent with the findings shown in Figure 5. This also indicated that SGF can significantly refine the grain structure of Cu-0.5Sn alloys.

By comparing the microstructures of the alloy grains obtained under normal and supergravity conditions through EBSD, as shown in Figure 7, it is obvious that the alloy grains were refined after the supergravity treatment. The inverse pole figure maps of the as-cast Cu-0.5Sn alloys obtained at G = 1 are shown in Figure 7a, which reveal that only part of the surface of an individual grain was available, and the whole grain could not be observed due to the large grain size. With the gravity coefficient increased to 500, fine grains can be found in the sample, as shown in Figure 7c. Moreover, sub-crystals of smaller grain sizes were formed when the gravity coefficient increased to G = 1000, as depicted in Figure 7d, which further demonstrates the role of supergravity in refining grain size.

Figure 8 depicts the strain–stress curves of as-cast Cu-0.5Sn samples acquired under supergravity conditions at room temperature. As the gravity coefficient increased, it is evident that the stress–strain curves became taller and broader, indicating increases in both strength and plasticity. The ultimate tensile strength value of the Cu-0.5Sn alloys was only 142.5 MPa in the normal gravity field, while the ultimate strength increased to 160.2 MPa in the SGF of G = 100. When the gravity coefficient increased to 1000, the strength reached 182 MPa. This is the distinguishing feature of grain refinement, as it increases the plasticity and toughness of the metal, while improving its strength. It can be explained by the Hall–Petch formula [4]:(2)$$\sigma =\sigma_{0}+kd^{-1/2}$$
where *d* is the average diameter of the grain; *k* is a constant. The finer the grains in polycrystalline, the greater their yield strength. During the deformation of polycrystalline, dislocations were blocked by and pinned to the grain boundaries, preventing the dislocations from sliding inside the grain boundaries, and the alloy was ultimately strengthened. At the same time, the grain refinement only causes an increase in the crystal interface, and the resulting lattice distortion caused was small, so it had a little effect on the conductivity of the material. The specific experimental results are listed in Table 2.

### 3.3. Effect Mechanism of Supergravity on the Grain Refinement

According to the solidification principle, the grain size depends on the ratio of the nucleation rate (N) to the growth rate (v). The larger the ratio of N/v is, the finer the grains are.

1.Ratio of the nucleation rate (N)

The metal solidification nucleation process can be divided into uniform and non-uniform nucleation [28]. The critical nucleation work, $\Delta G_{gG}^{*}$, and critical nucleation radius, $r_{gG}^{*}$, for uniform nucleation under the gravity field can be expressed by Equations (3) and (4), respectively:(3)$$\Delta G_{gG}^{*}=\frac{16}{3}\frac{\pi \sigma^{3}}{{\left(\Delta G_{V}+K_{g}+K_{G}\right)}^{2}}$$
(4)$$r_{gG}^{*}=\frac{2\sigma }{\Delta G_{V}+K_{g}+K_{G}}$$
where $\Delta G_{V}$ is the Gibbs free energy difference per unit volume of the solid–liquid, σ denotes the solid–liquid interfacial tension, $K_{g}$ and $K_{G}$ show the energy per volume exerted on the nucleus provided by normal gravity and supergravity, respectively. The values of $K_{g}$ and $K_{G}$ are positive, and $K_{G}\geq K_{g}$.

From Equation (3), it is evident that both increasing $K_{G}$ and increasing $K_{g}$ can reduce the critical nucleation work $\Delta G_{gG}^{*}$ of uniform nucleation and reduce the critical nucleation radius $r_{gG}^{*}$, which provides the possibility of a good solution for grain refinement under supergravity. Secondly, the intense convection induced by supergravity can also lead to the fracture of dendrites [26]. Due to the density difference between the dendrite fragments and the melt, they float in the direction of melt movement under the influence of SGF and become non-uniform nucleated particles. According to the expression of critical nucleation work, $\Delta G_{het}^{*}$, for heterogeneous nucleation (Equation (5)), these free grains and fragments came from another part of the sample with similar crystal structures (e.g., atomic arrays and interatomic distances), and the wetting angle, *θ*, must be close to 0. Therefore, the critical nucleation work for nucleation also tends to 0, which indicates that only a small degree of supercooling is required for heterogeneous nucleation caused by supergravity [26]. This promotes nuclei formation in advance and increases the ratio of the nucleation rate.
(5)$$\Delta G_{het}^{*}=\frac{16\pi \sigma^{3}}{3\Delta G_{V}^{2}}(\frac{2-3\cos\theta +\cos^{2}\theta }{4})$$

2.Growth rate (v)

The molten metal accelerates its motion due to it being subjected to extreme forces in the SGF. The pressure, *P* (Pa), acting on the molten metal caused by centrifugal force can be calculated by [30]: (6)$$P=\frac{\rho \omega^{2}(L_{2}^{2}-L_{1}^{2})}{2}=2\rho n^{2}\pi^{2}(L_{2}^{2}-L_{1}^{2})$$
where *ρ* denotes the molten Cu-0.5Sn alloy density, 8.920 g·cm^−3^ n is the rotational speed, with values of 10.0, 17.3, 22.3, and 31.5 r·s^−1^ corresponding to the supergravity coefficients of G = 100, 300, 500, and 1000, respectively. In this work, *L*_2_ and *L*_1_ represent the molten Cu-0.5Sn alloy levels as measured from the center of rotation, which are 0.27 m and 0.24 m, respectively.

Based on the aforementioned formula, the resultant pressure is calculated as 0, 269.4, 806.3, 1339.7, and 2673.1 kPa, corresponding to the supergravity coefficients of G = 1, 100, 300, 500, and 1000 in this paper. This tremendous pressure not only has an influence on grain refining, but it also changes the viscosity.

It is assumed that all the molten metal is converted into the ideal gas at the standard atmospheric pressure (*P*_0_) and constant boiling point ($T_{vap}$), and this is combined with the laws of thermodynamics to describe the following function [29]:(7)dU=TdS−pdV
(8)pV=nRT
(9)$$S_{0}=nC_{1}\ln T_{vap}+nR\ln P_{0}$$
where *R* is the gas constant, which is 8.314 J·mol^−1^·K^−1^, n is the moles of gas (mol), *C*_1_ is the molar heat capacity of the gas (J·mol^−1^·K^−1^), and *S*_0_ is the entropy value of an ideal alloy gas. After being transferred back into the molten metal at *T_e_* = 1200 °C, the system’s entropy decreased as follows:(10)S=−nRlnp−A
(11)$$A=S_{0}-nC_{2}\ln\frac{T_{vap}}{T_{e}}-\frac{\Delta H_{vap}}{T_{vap}}$$
where *A* is a complex constant, $\Delta H_{vap}$ is the latent heat of vaporization, and *C*_2_ is the molar heat capacity of the liquid. The viscosity (*η*) of the liquid changes with entropy, which can be expressed as $\eta =\eta_{0}\exp(\frac{c}{TS_{c}})$, and *c* is a constant. It can be concluded that
(12)$$\eta =\eta_{0}\exp(\frac{c}{T_{e}(-nR\ln p-A})$$

The pressure value of the melt increases from 0 to 2673.1 kPa, with the gravity coefficient increasing from 1 to 1000 through Equation (6). Therefore, the viscosity of the melt increases. Based on the Stokes–Einstein connection, this results in a lower diffusion coefficient of $D_{eff}$. So, the rate at which atoms travel toward the nucleus decreases, thus lowering the rate of grain growth.
(13)$$D_{eff}=\frac{kT}{3\pi \eta a}$$

In summary, the severe convection in the SGF causes the fracture of dendrites, which become new nucleation particles. At the same time, it promotes the formation of nuclei in advance and increases the ratio of the nucleation rate. Additionally, as the gravity coefficient increases, the increase in viscosity hinders the diffusion of atoms in the melt, leading to a decrease in the grain growth rate. Consequently, the larger the ratio of N/v is and the finer the grain size becomes with the increase in the gravity coefficient.

## 4. Conclusions

The influence of SGF on the microstructure and mechanical properties of high-conductive Cu alloys (as-cast Cu-0.5Sn alloys) was studied, and the observed conclusions are summarized as follows:It was experimentally verified that the addition of Sn could refine the grain, and the ultimate tensile strengths of the samples were enhanced to 110.8 MPa, 127.9 MPa, and 145.2 MPa, corresponding to 0wt%, 0.3wt%, and 0.5wt% of the Sn content, respectively. However, the conductivity of the alloy was significantly reduced.Supergravity can significantly refine the grain structure of Cu-0.5Sn alloy, and the average grain size was 0.8 mm at G = 1000. Meanwhile, the shape of the crystals gradually transformed from coarse crystals into equiaxed crystals as the gravity coefficient increased. Moreover, both the plasticity and tensile strength were enhanced with the increased gravity coefficient. As the gravity coefficient increases from 1 to 1000, the ultimate tensile strength of Cu-0.5Sn alloys increased from 145.2 MPa to 182 MPa, and there was almost no effect on conductivity.The mechanism of grain refinement by the SGF was clarified. Intense convection caused the fractured dendrites to become new nucleating particles, which promoted nuclei formation in advance and increased the ratio of the nucleation rate. It was also found that the melt pressure increased from 0 to 2673.1 kPa with the increase in the gravity coefficient from 1 to 1000. The increased viscosity under SGF hindered the diffusion of atoms in the melt and slowed down the movement of atoms toward the nucleus, which did not benefit from grain growth and led to a decrease in grain size.

## Figures and Tables

**Figure 1 materials-16-02485-f001:**
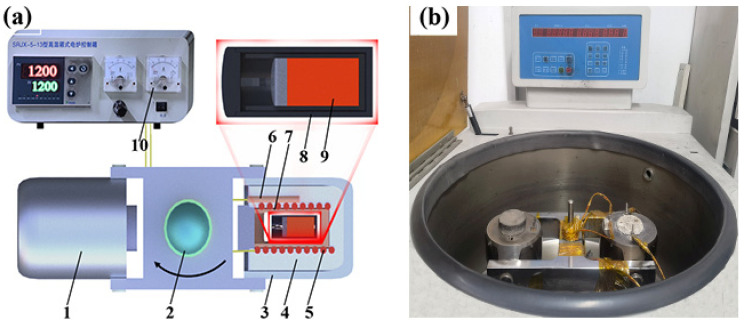
Schematic (**a**) and image (**b**) of the centrifugal apparatus: (1) counterweight; (2) centrifugal axis; (3) heating furnace; (4) refractory materials; (5) resistance coil; (6) thermocouple; (7) furnace chamber; (8) graphite and quartz crucible; (9) metal melts and covering slag; (10) temperature controller.

**Figure 2 materials-16-02485-f002:**
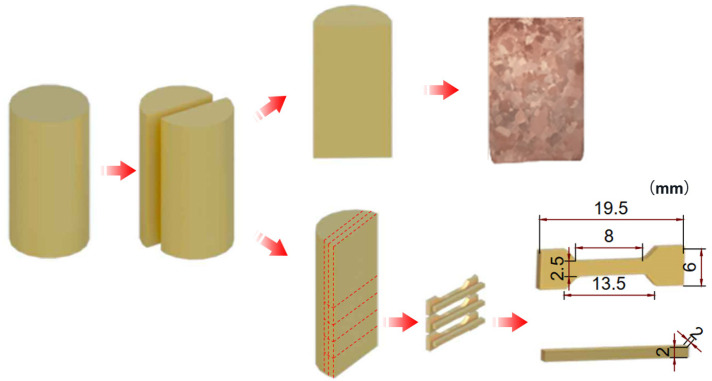
Schematic geometry of grain size characterization, electrical conductivity, and tensile testing specimen.

**Figure 3 materials-16-02485-f003:**
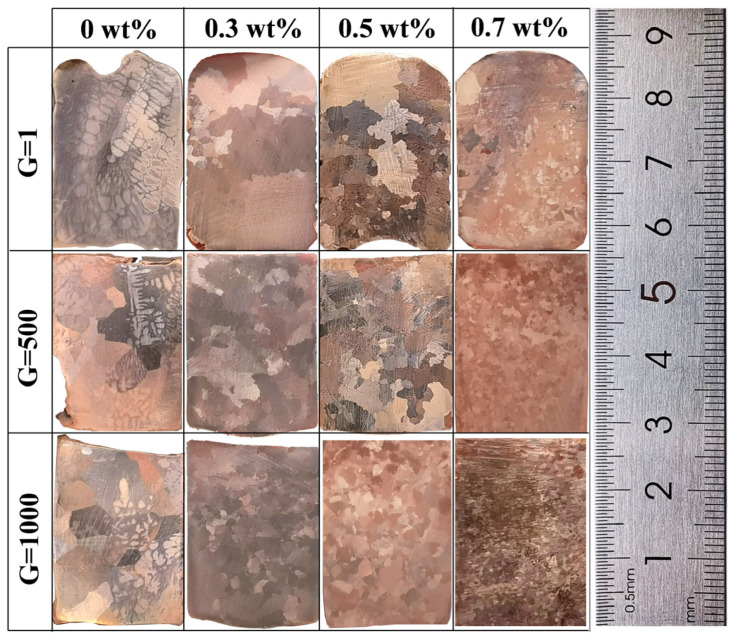
Effect of different Sn contents on the grain size of Cu alloy under different gravity coefficients.

**Figure 4 materials-16-02485-f004:**
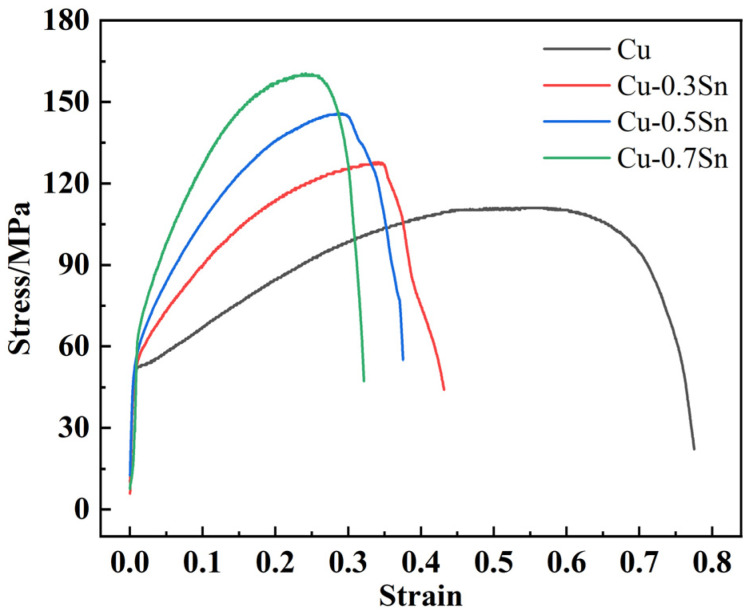
The stress–strain curves of Cu alloys with different Sn contents.

**Figure 5 materials-16-02485-f005:**
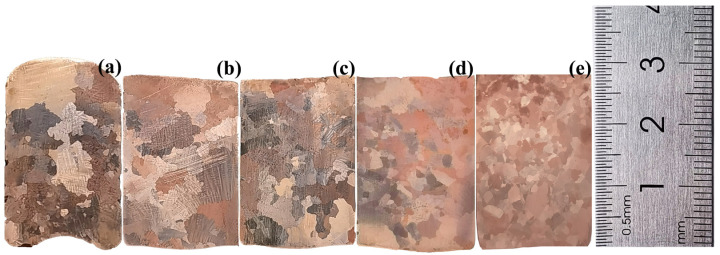
Effect of different gravity coefficients on grain size of Cu-0.5Sn alloy: (**a**) G = 1, (**b**) G = 100, (**c**) G = 300, (**d**) G = 500, and (**e**) G = 1000.

**Figure 6 materials-16-02485-f006:**
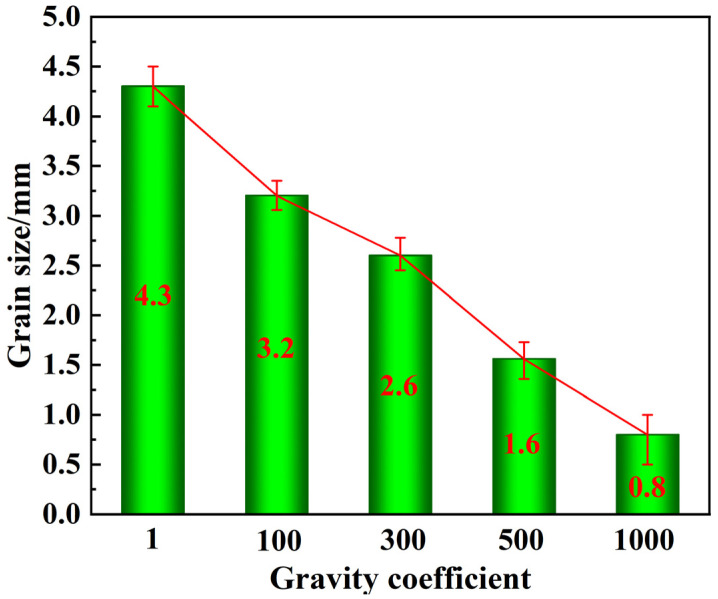
Average grain size of the Cu-0.5Sn alloy under different gravity coefficients.

**Figure 7 materials-16-02485-f007:**
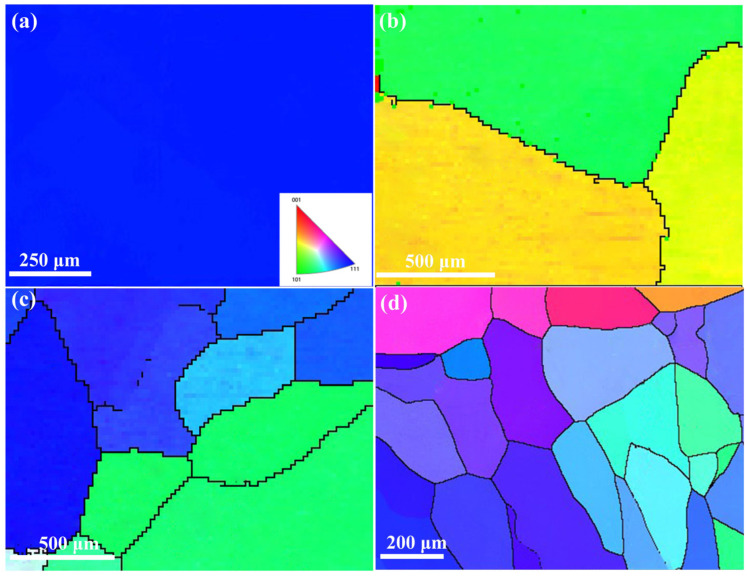
EBSD characterization of the as-cast Cu-0.5Sn alloys: inverse pole figure maps and corresponding grain size distributions of (**a**) G = 1, (**b**) G = 100, (**c**) G = 500, and (**d**) G = 1000.

**Figure 8 materials-16-02485-f008:**
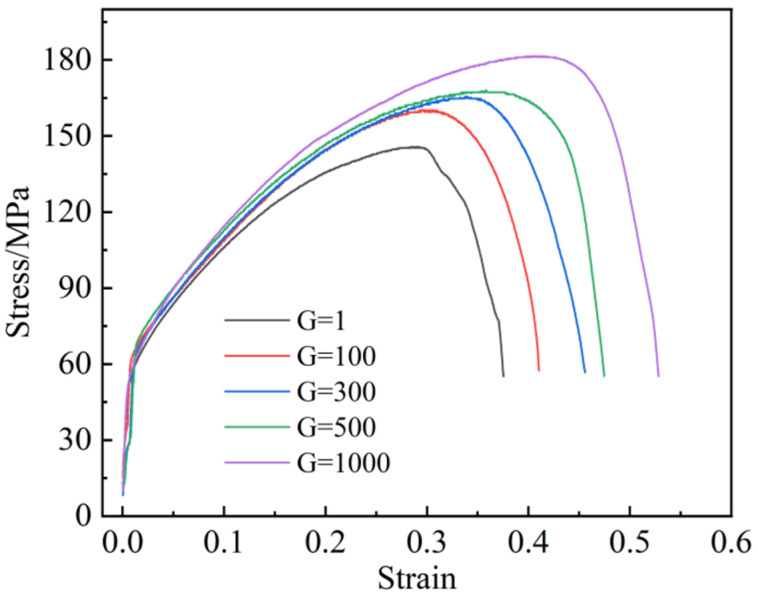
The stress–strain curves of Cu-0.5Sn alloys under different gravity coefficients.

**Table 1 materials-16-02485-t001:** The different Sn contents: experimental results.

	Conductivity (%IACS)	Standard Deviation	Ultimate Tensile Strength (MPa)	Standard Deviation
Cu	104	0.8	110.8	2.3
Cu-0.3Sn	94	1.5	127.9	1.2
Cu-0.5Sn	83	1.1	145.2	1.7
Cu-0.7Sn	73	1.2	160.6	2.7

**Table 2 materials-16-02485-t002:** The experimental results under different gravity coefficients.

Gravity Coefficient	Grain Size/mm	Conductivity (%IACS)	Standard Deviation	Ultimate Tensile Strength (MPa)	Standard Deviation
1	4.3	83	1.1	145.2	1.7
100	3.2	84	1.2	160.2	1.5
300	2.6	85	0.6	165.3	1.8
500	1.56	84	0.9	167.9	1.1
1000	0.8	83	0.3	182.0	1.3

## Data Availability

Data available in a publicly accessible repository.

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
