# Peer review of "Effect of Supergravity Field on the Microstructure and Mechanical Properties of Highly Conductive Cu Alloys"

_materials, 2023, doi:10.3390/ma16062485_

Round 1

Reviewer 1 Report

Review comments attached

Reviewer 2 Report

Review report

In the manuscript entitled “Effect of Super-Gravity Field on the Microstructure and Mechanical Properties of High Conductive Cu Alloys,” the authors attempt to utilize the super-gravity field (SGF) on the microstructure formation during the solidification in Cu-Sn alloys. And the authors also investigated those mechanical properties resulting in the microstructure. The authors carefully discussed the effect of the super-gravity field on nucleation, but the reviewer would like to comment as follows.

As aforementioned, the authors discuss the effect of the SGF based on the free energy difference. However, the authors are considering the only case of homogeneous nucleation. On the other hand, the authors mention the inhomogeneous nucleation just after. Especially in the case of inhomogeneous nucleation, no term relating to the SGF is included in the equation (5). To avoid confusing the reader, the authors should discuss the interrelation between homogeneous and inhomogeneous nucleation and the SGF effect more carefully. Also, the authors suggest discussing how we consider the nucleation mechanism in the solidification under the SGF (such as that is homogeneous? or inhomogeneous?).

Furthermore, the reviewer guesses that equation (9) is derived from the Gibbs free energy; however, the authors expressed the internal energy as the equation of dU. Strictly speaking, it is correct. But the authors discuss the nucleation as the Gibbs free energy; therefore, the reviewer suggests that the authors mention the derivation of eq.(9) starting from the Gibbs free energy or express them in more detail. Also, how did the authors transform equations (10) and (11)? The derivation from equations (7) to (12) would be more understandable to readers, even in an appendix or supplement.

Small comments

1. In L.196, the reviewer recommends adding the results of the other Cu-Sn alloys with different contents.

2. Concerning figure 6, is it possible to revise the function of the gravity coefficient in the horizontal axis, like 0 to 1000, to understand their functional relationship, like linear, quadratic, or the others?  

3. Concerning figure 7, the reviewer recommends describing what the authors obtain in “EBSD,” not data that can be understood in the micrograph.

4. There is no detailed description of equation (13) in the text; why did the authors describe it?

5. Regarding 1st conclusion,  is it possible to measure Sn distribution after solidification?

6. The authors must complete the parts of author contributions etc., after conclusions.

7. There are unclear expressions in the manuscript. For example, the reviewer considers the revised word is better. Not only the following comments, the reviewer firmly asks the authors to check and revise the English in the manuscript carefully and thoroughly.

7-1. flaws in L.34 → defects

7-2. Illustrate in L.166 → show, or represent. (because fig.5 is not an illustration.) 

7-3. fracture in L.181 etc. → breaking

7-4. work in L.235 is unclear what the authors want to say.

Reviewer 3 Report

1. The author include unit of super gravity field in abstract line 15.

2.The author should explain "it was experimentally verified that the addition of Sn  could effectively refined the grain" if there is any reason, and include microstructure analysis to find refined structure.

3. The author include more references related to your work for better understanding of beginners. 

4. SEM analysis is required.

5. Fractography images are required to understand the failure occurs in the specimen.

6. If there is any reason to select the gravity coefficient level (1 to 1000)

7. In figure 6, include the standard deviation level and values in the graph. 

8. What is the reason to decrease the level in figure 6.

9. In page no.6 line 185, "shape of crystals gradually transformed into equiaxed crystals". How? 

Round 2

Reviewer 2 Report

The authors thoroughly revised the manuscript with their reply. I recommend this manuscript for publication in "Materials" after the revision on the following points.

Concerning Fig.7, seeing the scale bar on each map is difficult. The reviewer suggests that it is revised as white colored or outline character.
